

# Lead and cadmium toxicity effects on the *Pinus roxburghii* seed germination and early seedling growth in different environments

Shabana Bibi[1], Tour Jan[1], Nasrullah Khan[1], Muhammad Wahab[2], Mohammad K. Okla[3], Bandar M. Almunqedhi[3], Ibrahim A. Saleh[4], Yasmeen A. Alwasel[3], Saud Alamri[3], Hayat Ullah[1] and Mostafa A. Abdel-Maksoud[3]

[1] Department of Botany, University of Malakand, Chakdara, Pakistan
[2] Department of Botany, Women University Swabi, Swabi, Pakistan
[3] Department of Botany and Microbiology, College of Science, King Saud University, Riyadh, Saudi Arabia
[4] Department of Medical Laboratory Sciences, Faculty of Allied Medical Sciences, Zarqa University, Zarqa, Jordan

Corresponding author
Tour Jan, tour_jan@yahoo.com

## ABSTRACT

Heavy metal toxicity affects germination, seedling growth, and other physiological processes in plants. To assess the toxic effects of heavy metals on the seed germination and seedling growth of *Pinus roxburghii*, we tested lead (Pb) and cadmium (Cd) at multiple concentrations ranging from 30–180 mg/L in both heterogeneous and homogeneous environments. The results showed that all seeds germinated regardless of Pb and Cd concentrations and environmental conditions, and the final germination percentage (GP) remained unchanged. Under different Pb and Cd stress levels, the seedlings grown under homogeneous conditions exhibited a higher stress tolerance index (STI) in morphological traits than those grown in heterogeneous environments. Pb and Cd also affected physiological indicators, their various concentrations promoted free proline in both environmental conditions. Sugar content in seedlings grown in heterogeneous environments ranged from 47.82% to 6.52% with Pb and 58.69 to 4.34% with Cd. In seedlings grown in homogenous environments, sugar content ranged from 45.71% to 5.71% with Pb and 37.14 to 2.85% with Cd. Chlorophyll a/b and carotenoids exhibited declining trends, with chlorophyll 'a' declining more than chlorophyll 'b'. This research indicates that *P. roxburghii* can be successfully used for plant restoration. It provides insights into potential hyper-tolerance mechanisms and can be utilized as a potential tree for roadside plantations to alleviate air pollution.

# INTRODUCTION

Lead (Pb) and cadmium (Cd) are two heavy metals with significant toxicity effects on all life forms, including flora (*Souri, Alipanahi & Tohidloo, 2016*). As non-essential elements, Pb and Cd can be readily absorbed by plant roots, leading to antagonistic effects on growth, development, and metabolism (*Marschner, 2011*). Plant species may react differently to

heavy metals, but Pb and Cd treatment generally decreases plant growth (*Chugh, Gupta & Sawhney, 1992*; *Villiers et al., 2011*). According to *Augusto et al. (2014)*, the impact of Cd on plants varies depending on the duration of exposure leading to morphological, physiological, and structural changes, as well as alterations in enzymatic and metabolic processes. The metabolic activities generate reactive oxygen species (ROS) in small amounts. In stressful environments, this ROS generation exceeds the maximum level (*Aprile et al., 2019*; *Huybrechts et al., 2019*), leading to oxidative stress (*Loix et al., 2017*; *Ali et al., 2020*). In soil, Cd significantly inhibits seed germination, growth, and other plant functions by creating free radicals and causing oxidative stress (*Shah et al., 2010*). It can damage the plant's root system (*Singh & Thakur, 2014*). Pb disturbs nutrient uptake through roots, alters membrane permeability, and disorders chloroplast ultrastructure, triggering changes in respiration and transpiration activities. It also creates ROS and activates some enzymatic and non enzymatic antioxidants (*Rahman et al., 2024*). Several physiological indicators, such as photosynthetic pigments, antioxidant enzymes, and antioxidants, can be utilized to assess the stress caused by heavy metals (*Singh, Nath & Sharma, 2007*). Heavy metals decrease carbon absorption, reformatting the chloroplast ultrastructure and thylakoid composition (*Hakmaoui et al., 2007*; *Pietrini et al., 2006*; *Aprile et al., 2019*). Seed germination is a particularly sensitive stage in the plant life cycle, crucial for crop production success, especially in dry and semi-arid regions (*Bezini et al., 2019*). Numerous studies have investigated the effects of heavy metals on seed germination (*Moosavi et al., 2012*). Examining the impacts of metal stress is vital for determining a species' potential to establish in metal-contaminated soils, as seed germination and the seedling stage are more susceptible to metal stress than later vegetative stages (*Bae, Benoit & Watson, 2016*). The common effects include a decreased germination rate and damage to seedlings' agronomic traits, such as a reduction in the lengthening and growth of roots, shoots, or leaves, potentially leading to seedlings' death (*Singh, Nath & Sharma, 2007*).

Phytotolerance studies are essential for understanding metal tolerance in plant species to comprehend the detrimental effect of metals on their metabolism and processes (*Jayasri & Suthindhiran, 2017*). In green remediation approaches, assessing various plant species' tolerance to contaminants and their accumulative behavior is crucial (*Nouri et al., 2017*). While most phytoremediation research occurs in greenhouses and labs, these conditions do not accurately represent remediation capacities in terrestrial ecosystems (*Rostami & Azhdarpoor, 2019*).

Assessments of success practiced in the nursery and in the field should be optimized to increase plant survival rates under stress conditions (*Grossnickle, 2012*). *Singh & Malaviya (2019)* noted that the removal efficiency of plants in real-world situations is influenced by multiple environmental factors, meaning that results obtained in controlled environments may differ significantly from those obtained in open fields. *Pinus eldarica*, *Wistaria sinensis*, *Morus alba*, and *Nigral morus* have the greatest bioaccumulation capacity for heavy metals (*Alahabadi et al., 2017*). Native plants are often preferred for phytoremediation due to their greater adaptation, survival, and development under environmental stress than non-native species (*Guarino et al., 2018*). *Pinus roxburghii* is a long lived tree; thus, the accumulated toxic elements can reside longer. The bark of *P. roxburghii* has been evaluated

for eliminating Chromium (*Ahmad, Rao & Masood, 2005*) and leaves for removing Arsenic from industrial wastewater (*Shafique et al., 2012*). *P. roxburghii*, the main Himalayan forest-forming tree, extends from Northern Pakistan, across Northern India and Nepal to Bhutan (*Puri et al., 2011*). *P. roxburghii* is an important species in the forest industry of northern Pakistan. It is extensively planted for timber in its native habitats and as an ornamental. The seeds of this species germinate quickly, making them valuable for forest nurseries focused on seedling establishment and reforestation efforts. Given its ecological needs, *P. roxburghii* can be effectively utilized to develop green spaces in high-traffic areas throughout northern Pakistan. Native plants are crucial for maintaining forest ecosystems and biodiversity because they adapt to their natural environment. The potential of forest trees for phytoremediation remains relatively unexplored. Conducting germination tests under varying concentrations of heavy metals can help identify new plant species with phytoremediation potential. This study focuses on evaluating the effects of Pb and Cd on the germination and seedling growth of *P. roxburghii* aiming to identify new native plants for use in phytoremediation efforts that can be effectively utilized in the future to develop green spaces in high traffic areas.

## MATERIALS AND METHODS

### Experimental setup

The experiments were conducted in two distinct environments: a heterogeneous (nursery) and a homogeneous (laboratory) setting. The open-air heterogeneous experiment occurred at the Bandagai Nursery in the Dir Lower district of Khyber Pakhtunkhwa, Pakistan, at 71.8195001°E longitude and 34.7460993°N latitude. Meanwhile, the homogeneous experiment was conducted under standard laboratory conditions at the Department of Botany, University of Malakand, Khyber Pakhtunkhwa, Pakistan. The experimental trials were commenced in April 2021. Pb and Cd were applied four times, at 12-day intervals, with equivalent water supplementation provided to control seedlings. After one and a half months of growth, the seedlings' photosynthetic and physiological characteristics were measured. The seedlings were harvested at the end of June 2021, with roots, shoots, and leaves harvested separately for morphological analysis.

### Germination assay

The same germination method was employed for nursery and laboratory experiments. *P. roxburghii* seeds were obtained from a local nursery. Seeds of uniform size were surface-sterilized for one minute using a 0.01% mercuric chloride ($HgCl_2$) solution before germination. Any residues of the hazardous solution were then thoroughly removed by washing with distilled water. Stock solutions of heavy metals—lead nitrate ($Pb(NO_3)_2$) and cadmium chloride ($CdCl_2$) were prepared in distilled water at concentrations of 0, 30, 60, 90, 120, 150, and 180 mg/L, respectively, based on a prior study by *Pavel et al. (2013)*. For the germination of *P. roxburghii* seeds, plastic pots (16 cm length and 8 cm diameter) filled with natural soil were utilized. Each treatment included three duplicates of nine seeds, which were watered with 100 mL of appropriate heavy metal solution ($Pb (NO_3)_2$ and $CdCl_2$) per pot, while controls received the same amount of distilled water. Germination

was recorded over 12 days, and the germination percentage (GP) was calculated using the following formula.

$$GP = \frac{Germinated\ Seed}{Total\ Seed} \times 100.$$

## Determination of morphological indices

The effects of Pb and Cd concentrations on the growth of seedlings were assessed under heterogeneous (nursery) and homogeneous (laboratory) conditions. After a four-week germination period, seven seedlings were randomly selected from each replicate to measure root, shoot, and leaf lengths, as well as to calculate the mean number of leaves, fresh and dry weights of the seedlings.

Root and shoot lengths were measured using a ruler. In the laboratory, the seedlings were initially placed in an oven at 100 °C for 30 min, followed by drying at 70 °C until a constant weight was achieved before being weighed.

## Stress Tolerance Index

This step aimed to determine the tolerance threshold of (*P. roxburghii*) to varying levels of Pb and Cd. We measured and weighed both metal-treated seedlings and control seedlings, which were grown in a nursery and under laboratory conditions. Plant materials were initially dried in an oven at 100 °C for 30 min to assess their dry biomass. Subsequently, they were further dried at 70 °C until they reached a constant weight and then weighed. The stress tolerance index (STI) (%) for the seedling was computed using the following formula based on the dry biomass:

$$STI = \frac{Dry\ weight\ of\ seedling\ developed\ on\ media\ with\ Pb(NO_3)_2\ or\ CdCl_2}{Dry\ weight\ of\ seedling\ developed\ on\ media\ devoid\ of\ Pb\ (NO_3)_2\ or\ CdCl_2} \times 100.$$

## Determination of the biochemical indices

From the three replicates of each treatment, three mature fresh leaves were indiscriminately selected from both heterogeneous and homogeneous grown seedlings and placed separately into an ice box to measure each biochemical index.

## Proline

Fresh needles (0.2 g) were homogenized in 10 mL of 3% sulfosalicylic acid solution in test tubes and then centrifuged at 10,000 rpm for 6 min to adjust the supernatant. Equal volumes (2 mL each) of the supernatant, ninhydrin reagent, and glacial acetic acid were mixed in the test tube. The solution was then shaken and heated in the water bath at 100 °C for 1 hour. After heating, the solution was cooled, and 4 mL of toluene was added to each test tube (*Bates, Waldren & Teare, 1973*), with toluene serving as a blank sample. The absorbance of the supernatant was measured using a UV spectrophotometer at 520 nm. The proline concentration was determined using a proline standard curve.

## Sugar

The sugar content in the needles of *P. roxburghii* (Sarg.) was determined using the phenol sulphuric acid method described by *Dubois et al. (1956)*. To each test tube, 10 mL of 80%

methanol and 0.2 g of needle samples were added. The solution was heated in a water bath at 95 °C for 10 min and then cooled to room temperature. Subsequently, 0.5 mL of the sample was mixed with 1 mL of 18% phenol and incubated at room temperature for 1 hr. Then, 2.5 mL of $H_2SO_4$ was supplemented, and the mixture was vortexed for 1 min. The absorbance of the sugar content was measured at 490 nm using a spectrophotometer in one mM cuvettes against the blank.

## Photosynthetic pigments

One gram of fresh pine needles was cut into small fragments and ground with a mortar and pestle to determine chlorophyll content. Then, 20 ml of 80% acetone and 0.5 g of $MgCO_3$ powder were added, and the mixture was ground further according to the method of *Kamble et al. (2015)*. The mixture was incubated at 4 °C for 3 hrs. After incubation, the mixture was centrifuged at 1,000 rpm for 10 min. The supernatant solution was transferred to a 100 mL volumetric flask and diluted to 100 ml with 80% acetone. The absorbance of the supernatant was measured for chlorophyll a/b at 663 nm and 645 nm using a spectrophotometer, with 80% acetone used as a blank sample. Chlorophyll content was determined using the appropriate formula given below:

$$\text{Chlorophyll a} = \frac{12.7 \ (A663) - 2.69 \ (A645) \times V}{1000 \times W}$$

$$\text{Chlorophyll b} = \frac{22.9 \ (A645) - 4.68 \ (A663) \times V}{1000 \times W}$$

$$\text{Total chlorophyll} = \frac{20.2 \ (A645) + 8,02 \ (A663) \times V}{1000 \times W}.$$

## Carotenoid

The carotenoid content was estimated following the method described by *Lichtenthaler (1987)*. Needles of *P. roxburghii* (Sarg.) were cut and extracted with 100% acetone using a pistil and mortar. The pigment extracts were centrifuged in glass tubes for 3 to 5 min. The resulting extracts were immediately analyzed using spectrophotometry.

## Data analysis

The study employed a completely randomized design (CRD) for data collection to ensure unbiased distribution of treatments across experimental units. The collected data were analyzed using one-way analysis of variance (ANOVA), a robust statistical method that assesses significant differences among group means. To further explore pairwise comparisons between treatment groups, a post-hoc test was conducted, which helps identify specific groups that differ significantly from each other. Pearson correlation coefficients were calculated to determine the strength and direction of relationships between variables, specifically assessing the effects of Pb and Cd treatments. The statistical analyses, including ANOVA with *post-hoc* tests and correlation assessments, were performed using OriginLab version 24.0b (OriginLab, Northampton, MA, USA), a powerful software for data visualization and statistical analysis. The significance level was set at $P < 0.05$, ensuring that only results with a probability of occurrence of less than 5% were considered statistically significant.

## RESULTS

### Germination assessment in the heterogeneous and homogeneous environment

The Pb and Cd treatments did not significantly affect the final seed germination rate, as all seeds germinated, regardless of the environmental conditions or the concentration of Pb and Cd. The only noticeable impact of these metal treatments was a negligible delay in seed germination compared to the control (Table 1). At higher concentrations (180 mg/L), germination began 5–6 days after seeding, whereas in the control, control group, germination started 3–4 days after seeding.

### Growth assessment in a heterogeneous environment

Pb and Cd considerably inhibited the morphological traits of *P. roxburghii* seedlings at various concentrations (30–180 mg/L). At 180 mg/L of Pb, root, shoot, and leaf lengths decreased by 55.40%, 72.24%, and 42.71%, respectively. At 30 mg/L of Pb, these traits were reduced by 17.82%, 18.94%, and 3.64%, respectively. Pb treatment had no significant effect on the number of leaves at concentrations ranging from 30 to 90 mg/L, though a slight reduction in leaf number was observed at concentrations ranging from 90 to 180 mg/L. Pb also substantially decreased the seedling's fresh and dry weights (Table 1). Likewise, Cd exposure significantly ($P < 0.05$) reduced morphological traits. At 180 mg/L, root, shoot, and leaf lengths were decreased by 44.93%, 70.04%, and 41.42%, respectively, while total fresh biomass decreased by 57.81% and dry biomass by 70.18%. A similar decreasing tendency in morphological traits was also observed at 30 mg/L of Cd (Table 1). Cd concentrations did not significantly affect the number of leaves, with effects comparable to the control.

### Biochemical assessment in heterogeneous conditions

Proline content initially increased and then decreased with the application of Pb and Cd compared to the control. The increase in proline ranged from 28.57% to 51.42% with Pb 30 to 180 mg/L, and 48.57% to 65.71% with Cd 30 to 180 mg/L (Table 2). Regarding sugar content, seedlings treated with low levels of Pb and Cd had higher sugar concentrations than those treated with high levels (Table 2). The content of photosynthetic pigments was also affected by Pd and Cd. Chlorophyll content in the treated seedling decreased progressively with increasing levels of Pb and Cd (Table 2). Total chlorophyll content decreased by 15.55 to 78.51%, and carotenoid dropped to 30.56 to 83.41% with Pb (30–180 mg/L). With Cd (30–180 mg/L), total chlorophyll content dropped from 25.37 to 77.41%, and carotenoid content decreased by 16.06 to 68.91% (Table 2).

### Growth assessment in homogeneous conditions

The impact of various Pb and Cd concentrations on the growth of *P. roxburghii* seedlings is detailed in Table 3. Both Pb and Cd positively enhanced morphological traits at a concentration of 30 mg/L (Table 3). However, at concentrations above 30 mg/L, the morphological traits showed only slight variations compared to the control (Table 3). The Pb and Cd most pronounced phytotoxic effects were observed at levels (180 mg/L),

**Table 1 Impact of Pb and Cd levels on germination and seedling growth under heterogeneous conditions.**

| Metals (mg/L) | | GP (%) | RL (cm) | SL (cm) | LL (cm) | LN | Total biomass (g) | |
|---|---|---|---|---|---|---|---|---|
| Pb | Cd | | | | | | FW | DW |
| Control | | 74.55 | 11.84 ± 0.05[a] | 4.54 ± 1.28[a] | 5.48 ± 0.13[a] | 12.42 ± 1.51[a] | 0.64 ± 0.25[a] | 0.493 ± 0.04[a] |
| 30 | – | 74.53 | 9.73 ± 0.73[ab] | 3.68 ± 0.55[ab] | 5.28 ± 0.03[a] | 12.32 ± 0.44[a] | 0.62 ± 0.34[ab] | 0.413 ± 0.33[ab] |
| 60 | – | 74.55 | 9.46 ± 1.39[ab] | 3.38 ± 0.21[b] | 4.56 ± 0.05[b] | 12.32 ± 0.30[a] | 0.32 ± 0.17[b] | 0.365 ± 0.23[c] |
| 90 | – | 74.55 | 8.48 ± 1.50[bc] | 3.14 ± 0.22b[c] | 4.13 ± 0.06[c] | 11.82 ± 0.74[a] | 0.28 ± 0.10[b] | 0.314 ± 0.09[d] |
| 120 | – | 74.54 | 8.06 ± 1.47b[cd] | 2.38 ± 0.29[c] | 3.68 ± 0.32[d] | 11.73 ± 0.19[a] | 0.16 ± 0.18[c] | 0.263 ± 0.06[e] |
| 150 | – | 74.53 | 7.64 ± 0.63[cd] | 1.54 ± 0.19[de] | 3.27 ± 0.05[e] | 11.52 ± 0.54[a] | 0.12 ± 0.07[de] | 0.213 ± 0.09[ef] |
| 180 | – | 74.53 | 5.28 ± 1.25[d] | 1.26 ± 0.65[e] | 3.14 ± 0.10[e] | 10.75 ± 0.30[ab] | 0.11 ± 0.10[de] | 0.179 ± 0.07[f] |
| *F-value* | | | 5.43 | 15.34 | 109.15 | 38.30 [ns] | 4.62 | 0.72 |
| *CV%* | | | 22.16 | 18.36 | 3.66 | 22.34 | 56.68 | 71.54 |
| – | 30 | 74.54 | 11.30 ± 0.10[b] | 4.07 ± 1.89[ab] | 5.42 ± 0.56[a] | 12.41 ± 0.6[a] | 0.49 ± 0.31[b] | 0.428 ± 0.32[ab] |
| – | 60 | 74.55 | 10.83 ± 0.15[c] | 3.66 ± 1.16[ab] | 5.18 ± 1.12[a] | 12.34 ± 0.63[a] | 0.46 ± 0.15[b] | 0.385 ± 0.30[ab] |
| – | 90 | 74.52 | 9.56 ± 0.41[d] | 3.24 ± 0.69[abc] | 4.52 ± 1.28[ab] | 11.40 ± 0.74[a] | 0.41 ± 0.34[ab] | 0.352 ± 0.29[b] |
| – | 120 | 74.53 | 8.89 ± 0.08[e] | 2.53 ± 0.59[abc] | 3.78 ± 1.36[b] | 10.80 ± 0.2[b] | 0.37 ± 0.46[c] | 0.287 ± 0.14[de] |
| – | 150 | 74.54 | 8.76 ± 0.24[e] | 2.16 ± 0.81[bc] | 3.42 ± 0.56[b] | 10.60 ± 0.81[b] | 0.35 ± 0.27[c] | 0.224 ± 0.09[c] |
| – | 180 | 74.54 | 6.52 ± 0.40[f] | 1.36 ± 0.72[c] | 3.21 ± 0.83[c] | 10.40 ± 0.67[b] | 0.27 ± 0.20[dc] | 0.147 ± 0.05[f] |
| *F-value* | | | 5.05 | 3.89 | 2.91 | 17.37 | 5.20 | 10.20 |
| *CV%* | | | 16.49 | 15.81 | 36.53 | 25.78 | 36.04 | 25.08 |

**Notes.**

Key: RL, Root length; SL, Shoot length; LL, Leaf length; LN, Leaf number; FW, Fresh weight; DW, Dry weight.

Means followed by different letters indicate a significant difference ($p < 0.05$) among different treatments according to the Tukey test.

compared to lower levels (60 mg/L). Specifically, Pb at 180 mg/L caused a 28.26% decline in root length, a 55.47% decline in shoot length, a 32.03% decline in leaf length, a 38.80% decline in total fresh biomass, and a 55.76% decline in total dry biomass. Similarly, Cd at 180 mg/L resulted in a 31.25% decrease in root length, a 39.67% decrease in shoot, a 35.03% decrease in leaf, a 22.38% decrease in fresh biomass, and a 53.20% decrease in dry biomass. At a concentration of 60 mg/L, Pb caused a 3.26% reduction in root length, 11.76% in shoot length, 10.82% in leaf length, 17.91% in fresh biomass, and 8.01% in dry biomass. Cd at 60 mg/L induced a 7.03% decrease in root length, 8.15% in shoot length, 6.01% in leaf length, 2.98% in fresh biomass, and 10.57% in dry biomass. Notably, the number of leaves was not significantly inhibited by Pb and Cd concentrations and remained comparable to the control.

**Biochemical assessment in homogeneous conditions**

The results presented in Table 4 show that exposure to 30–180 mg/L of Pb resulted in a 13.04–39.13% increase in proline content, while Cd led to a 43.47–26.08% increase (Table 4). Additionally, the sugar content in treated seedlings increases by 22.85% to 34.28% with Pb (30–120 mg/L) and by 17.14% to 28.57% with Cd (30–120 mg/L). At higher concentrations of Pb and Cd (150 and 180 mg/L), sugar levels returned to values similar to the control (Table 4). The treated seedling also showed a decreasing tendency in photosynthetic pigments across all levels of Pb and CD. Total chlorophyll decreased by 17.56–81.53% with Pb (30–180 mg/L), and by 14.41–80.63% with Cd (30–180 mg/L),

**Table 2  Effects of Pb and Cd concentrations physiochemical parameters under heterogeneous environments.**

| Heavy metal (mg/L) | | Proline $\mu$mol g$^{-1}$ | Sugar $\mu$g$^{-1}$ | Photosynthetic pigments (mg g$^{-1}$) | | | Carotenoid mg g$^{-1}$ |
|---|---|---|---|---|---|---|---|
| Pb | Cd | | | Chl a | Chl b | Total Chl (a+b) | |
| Control | | $0.35 \pm 0.07^d$ | $0.46 \pm 0.06^c$ | $2.73 \pm 0.80^a$ | $2.67 \pm 0.08^a$ | $5.40 \pm 0.88^a$ | $1.93 \pm 0.74^a$ |
| 30 | – | $0.45 \pm 0.05^c$ | $0.63 \pm 0.09^{ab}$ | $2.61 \pm 0.80^a$ | $1.83 \pm 0.06^b$ | $4.44 \pm 0.86^{ab}$ | $1.64 \pm 0.64^{ab}$ |
| 60 | – | $0.48 \pm 0.07^d$ | $0.68 \pm 0.13^a$ | $2.32 \pm 0.33^{ab}$ | $1.67 \pm 0.08^c$ | $4.0 \pm 0.33^{bc}$ | $1.23 \pm 0.32^{bc}$ |
| 90 | – | $0.49 \pm 0.05^{cbd}$ | $0.59 \pm 0.09^{abc}$ | $2.03 \pm 1.01^{abc}$ | $1.42 \pm 0.05^d$ | $3.45 \pm 1.06^c$ | $0.87 \pm 0.11^{cd}$ |
| 120 | – | $0.51 \pm 0.17^b$ | $0.56 \pm 0.15^{abc}$ | $1.33 \pm 0.82^{bcd}$ | $0.87 \pm 0.15^e$ | $2.2 \pm 0.97^d$ | $0.68 \pm 0.07^{cd}$ |
| 150 | – | $0.53 \pm 0.06^a$ | $0.52 \pm 0.07^{abc}$ | $1.16 \pm 0.28^{cd}$ | $0.57 \pm 0.11^f$ | $1.73 \pm 0.39^{de}$ | $0.52 \pm 0.08^d$ |
| 180 | – | $0.53 \pm 0.02^a$ | $0.49 \pm 0.03^{bc}$ | $0.82 \pm 0.15^d$ | $0.34 \pm 0.04^g$ | $1.16 \pm 0.19^e$ | $0.32 \pm 0.04^d$ |
| F-value | | 19.33 | 3.97 | 4.81 | 3.10 | 1.19 | 2.69 |
| CV-Value | | 2.34 | 38.56 | 17.19 | 6.97 | 57.44 | 61.63 |
| – | 30 | $0.52 \pm 0.12^a$ | $0.68 \pm 0.12^a$ | $2.36 \pm 1.57^{ab}$ | $1.67 \pm 0.07^b$ | $4.03 \pm 1.64^b$ | $1.62 \pm 0.63^{ab}$ |
| – | 60 | $0.52 \pm 0.03^a$ | $0.73 \pm 0.10^a$ | $2.13 \pm 0.22^{ab}$ | $1.47 \pm 0.21^{bc}$ | $3.60 \pm 0.43^{bc}$ | $0.96 \pm 0.15^{bc}$ |
| – | 90 | $0.54 \pm 0.10a$ | $0.65 \pm 0.08^a$ | $1.54 \pm 0.54^{bc}$ | $1.36 \pm 0.81^{bc}$ | $2.90 \pm 1.35^{bcd}$ | $0.87 \pm 0.14^c$ |
| – | 120 | $0.56 \pm 0.31^{ab}$ | $0.57 \pm 0.11^{abc}$ | $1.41 \pm 0.12^{bc}$ | $0.96 \pm 0.15^{cd}$ | $2.37 \pm 0.27^{cde}$ | $0.76 \pm 0.20^{dc}$ |
| – | 150 | $0.57 \pm 0.64^b$ | $0.52 \pm 0.03^{abc}$ | $1.33 \pm 0.57^{bc}$ | $0.53 \pm 0.03^{de}$ | $1.86 \pm 0.60^{de}$ | $0.71 \pm 0.24^{de}$ |
| – | 180 | $0.58 \pm 0.24^b$ | $0.48 \pm 0.01^{bc}$ | $0.94 \pm 0.14^c$ | $0.28 \pm 0.10^e$ | $1.22 \pm 0.24^e$ | $0.60 \pm 0.20^e$ |
| F-value | | 20.05 | 2.23ns | 5.58 | 32.34 | 25.97 | 52.64 |
| CV-value | | 12.34 | 15.82 | 30.20 | 5.90 | 16.51 | 32.02 |

Notes.
Chl a, chlorophyll a; Chl b, chlorophyll b.

as shown in Table 4. Carotenoid levels in *P. roxburghii* seedlings decreased by 33.99–71.14% and 32.01–69.96% under Pb and Cd at 30–120 mg/L, respectively. The respective values of carotenoids decreased by 77.86–85.37% and 77.07–83.39% with Pb and Cd at 150–180 mg/L, respectively Table 4).

## Stress tolerance index

In experiments with *P. roxburghii* seedlings in a heterogeneous environment, the STI exhibited a negative correlation with increasing concentrations of Pb and Cd. Specifically, STI values for Pb ranged from 83.77% to 36.30%, while for Cd, they ranged from 86.81% to 29.84% at concentrations of 30–180 mg/L (Table 5). Conversely, in a homogeneous environment, at a low concentration of 30 mg/L, the STI for Pb was 109%, whereas, for Cd, it was 112%, showing a positive correlation between STI and the concentrations of Pb and Cd at this low level. However, at higher concentrations (above 30 mg/L), the seedlings showed the lowest STI values, reaffirming a negative correlation between STI and the concentrations of Pb and Cd (Table 5).

## Correlation analysis

The correlation between various morphological traits and organic solutes in *P. roxburghii* seedlings exposed to Pb and Cd under both field and laboratory conditions is shown in Figs. 1 and 2, respectively. Our findings reveal that while not all morphological characteristics were significantly correlated, the majority exhibited strong correlations in both heterogeneous and homogeneous conditions. For example, stem length (SL)

**Table 3  Effect of different concentrations of Pb and Cd on seed germination and seedling growth under homogeneous conditions.**

| Heavy metals (mg/L) | | GP (%) | RL (cm) | SL (cm) | LL (cm) | LN | Total biomass (g) | |
|---|---|---|---|---|---|---|---|---|
| Pb | Cd | | | | | | FW | DW |
| Control | | 73.54 | $8.03 \pm 1.15^a$ | $8.36 \pm 1.02^{ab}$ | $6.65 \pm 0.65^{ab}$ | $11.4 \pm 1.52^a$ | $0.67 \pm 0.15^{ab}$ | $0.312 \pm 0.11^b$ |
| 30 | | 73.82 | $8.67 \pm 0.98^{ab}$ | $8.92 \pm 0.99^a$ | $6.84 \pm 1.25^a$ | $11.61 \pm 1.52^a$ | $0.69 \pm 0.38^a$ | $0.342 \pm 0.24^a$ |
| 60 | | 73.50 | $7.12 \pm 0.94^{bc}$ | $7.64 \pm 0.95^{ab}$ | $5.93 \pm 1.10^{abc}$ | $11.4 \pm 1.52^a$ | $0.55 \pm 0.35^c$ | $0.287 \pm 0.25^c$ |
| 90 | | 73.48 | $6.53 \pm 1.33^{bc}$ | $7.43 \pm 1.68^{abc}$ | $5.53 \pm 0.56^{abc}$ | $11.4 \pm 2.08^a$ | $0.52 \pm 0.42^c$ | $0.242 \pm 0.12^d$ |
| 120 | | 73.56 | $6.16 \pm 0.94^c$ | $6.32 \pm 1.19^{bc}$ | $5.23 \pm 0.77^{bc}$ | $11.43 \pm 1.15^a$ | $0.47 \pm 0.24^d$ | $0.209 \pm 0.10^e$ |
| 150 | | 73.53 | $5.64 \pm 0.64^c$ | $5.46 \pm 1.46^{bc}$ | $4.87 \pm 0.85^c$ | $11.4 \pm 1.52^a$ | $0.44 \pm 0.22^{de}$ | $0.176 \pm 0.07^f$ |
| 180 | | 73.51 | $3.86 \pm 0.05^d$ | $5.28 \pm 1.07^c$ | $4.52 \pm 1.76^c$ | $11.4 \pm 1.52^a$ | $0.41 \pm 0.10^{de}$ | $0.138 \pm 0.03^{fg}$ |
| *F-value* | | | *10.49* | *1.24* | *2.15* | *0.63ns* | *1.25* | *1.41* |
| *CV %* | | | *10.50* | *19.59* | *10.64* | *4.97* | *42.55* | *42.85* |
| | 30 | 73.53 | $8.12 \pm 0.88^a$ | $7.91 \pm 1.65^a$ | $7.01 \pm 1.00^a$ | $11.63 \pm 2.08^a$ | $0.69 \pm 0.09^a$ | $0.351 \pm 0.18^a$ |
| | 60 | 73.51 | $7.96 \pm 2.03^a$ | $7.23 \pm 0.90^{ab}$ | $6.25 \pm 0.76^{abc}$ | $11.4 \pm 1.00^a$ | $0.65 \pm 0.05^{ab}$ | $0.279 \pm 0.15^b$ |
| | 90 | 73.53 | $7.85 \pm 0.48^{ab}$ | $6.76 \pm 1.76^b$ | $5.73 \pm 0.73^{abc}$ | $11.4 \pm 0.57^a$ | $0.63 \pm 0.58^{ab}$ | $0.254 \pm 0.13^{bc}$ |
| | 120 | 73.50 | $6.23 \pm 0.79^{bc}$ | $6.56 \pm 2.32^d$ | $5.26 \pm 0.89^{bc}$ | $11.4 \pm 1.52^a$ | $0.61 \pm 0.54^c$ | $0.216 \pm 0.02^d$ |
| | 150 | 73.51 | $5.63 \pm 1.26^{cd}$ | $6.42 \pm 0.86^{de}$ | $4.78 \pm 1.54^{bc}$ | $11.4 \pm 1.15^a$ | $0.56 \pm 0.20^d$ | $0.178 \pm 0.06^e$ |
| | 180 | 73.52 | $5.06 \pm 1.02^d$ | $5.23 \pm 1.23^f$ | $4.32 \pm 2.17^c$ | $11.4 \pm 2.08^a$ | $0.52 \pm 0.22^d$ | $0.146 \pm 0.05^{ef}$ |
| *F-value* | | | *10.59* | *3.27* | *3.38* | *1.00ns* | *1.01* | *0.43* |
| *CV %* | | | *12.90* | *18.02* | *14.55* | *1.92* | *52.57* | *52.89* |

Notes.

Key: RL, Root length; SL, Shoot length; LL, Leaf length; LN, Leaf number; FW, Fresh weight; DW, Dry weight.

Means followed by different letters indicate a significant difference ($p < 0.05$) among different treatments according to the Tukey test.

under Pb exposure in laboratory conditions was strongly correlated with root length (RL), leaf length ($r = 0.695$, $P < 0.001$), and leaf number ($r = 0.714$, $P < 0.001$). However, under homogeneous conditions, only root length ($r = 0.60$, $P < 0.01$) and leaf length ($r = 0.576$, $P < 0.01$) showed a strong correlation with root length, while leaf number and the fresh and dry weight of seedlings did not show significant correlations, except for leaf length ($r = 0.466$, $P < 0.05$) and leaf number ($r = 0.446$, $P < 0.05$), respectively.

Similarly, the intercorrelation among morphological traits with Cd treatment was generally stronger than with lead (Fig. 2). However, like in the Pb-treated seedlings, leaf number, fresh weight, and dry weight were not correlated with any other morphological variables. In terms of organic solutes, those extracted from *P. roxburghii* seedlings in both field and laboratory conditions showed some significant relationships (Fig. 2). For instance, chlorophyll content (a and b) and carotenoid levels were significantly correlated with stem and leaf length in Pb-treated seedlings. In contrast, these relationships were even stronger in seedlings treated with Cd (Fig. 2).

## DISCUSSION

Our research paper investigates the effect of Pb and Cd on seed germination and seedling growth of *P. roxburghii* (Sarg) under heterogeneous and homogeneous conditions. While existing literature highlights the detrimental effects of heavy metals on overall plant growth, our study focused on evaluating the resistance of the key propagative organs, seeds, and

**Table 4  Effect of different concentrations of Pb and Cd physiochemical parameters under a homogeneous condition.**

| Heavy metal (m/L) | | Proline μmol g$^{-1}$ | Sugar μg$^{-1}$ | Photosynthetic pigments (mg g$^{-1}$) | | | Carotenoid mg g$^{-1}$ |
|---|---|---|---|---|---|---|---|
| Pb | Cd | | | Chl 'a' | Chl 'b' | Total Chl (a+b) | |
| Control | | 0.23 ± 0.05$^b$ | 0.35 ± 0.07$^d$ | 2.60 ± 0.55$^a$ | 1.81 ± 0.22$^a$ | 4.41 ± 0.77$^a$ | 2.53 ± 0.11$^a$ |
| 30 | – | 0.32 ± 0.05$^{ab}$ | 0.47 ± 0.02$^{ab}$ | 2.25 ± 0.05$^b$ | 1.41 ± 0.12$^b$ | 3.66 ± 0.17$^b$ | 1.67 ± 0.07$^b$ |
| 60 | – | 0.35 ± 0.05$^a$ | 0.51 ± 0.09$^a$ | 1.63 ± 0.06$^c$ | 1.02 ± 0.05$^c$ | 2.65 ± 0.11$^c$ | 1.33 ± 0.11$^c$ |
| 90 | – | 0.36 ± 0.06$^{ab}$ | 0.46 ± 0.06$^{abc}$ | 1.30 ± 0.09$^d$ | 0.83 ± 0.22$^c$ | 2.13 ± 0.31$^d$ | 0.96 ± 0.05$^d$ |
| 120 | – | 0.36 ± 0.05$^{ab}$ | 0.43 ± 0.07$^{abcd}$ | 0.85 ± 0.39$^e$ | 0.59 ± 0.10$^d$ | 1.44 ± 0.49$^e$ | 0.73 ± 0.23$^e$ |
| 150 | – | 0.34 ± 0.04$^a$ | 0.39 ± 0.04$^{bcd}$ | 0.67 ± 0.07$^e$ | 0.43 ± 0.07$^{de}$ | 1.10 ± 0.14$^{ef}$ | 0.56 ± 0.05$^{ef}$ |
| 180 | – | 0.32 ± 0.04$^{ab}$ | 0.37 ± 0.07$^{cd}$ | 0.46 ± 0.06$^e$ | 0.36 ± 0.04$^e$ | 0.82 ± 0.10$^f$ | 0.37 ± 0.07$^f$ |
| F-value | | 5.84 | 3.97 | 148.76 | 91.24 | 214.40 | 262.26 |
| CV-Value | | 15.90 | 10.67 | 6.93 | 10.33 | 6.09 | 6.80 |
| – | 30 | 0.33 ± 0.04$^a$ | 0.45 ± 0.05$^{ab}$ | 2.34 ± 0.04$^b$ | 1.46 ± 0.16$^b$ | 3.80 ± 0.20$^b$ | 1.72 ± 0.06$^b$ |
| – | 60 | 0.36 ± 0.05$^a$ | 0.48 ± 0.05$^a$ | 2.03 ± 0.07$^c$ | 1.35 ± 0.09$^b$ | 3.38 ± 0.17$^c$ | 1.18 ± 0.17$^c$ |
| – | 90 | 0.30 ± 0.04$^{ab}$ | 0.43 ± 0.06$^{ab}$ | 1.65 ± 0.05$^d$ | 0.74 ± 0.11$^c$ | 2.39 ± 0.16$^d$ | 0.94 ± 0.06$^d$ |
| – | 120 | 0.28 ± 0.03$^{bc}$ | 0.41 ± 0.05$^{abc}$ | 1.23 ± 0.25$^e$ | 0.63 ± 0.06$^{cd}$ | 1.86 ± 0.31$^e$ | 0.76 ± 0.06$^e$ |
| – | 150 | 0.29 ± 0.09$^c$ | 0.38 ± 0.02$^{bc}$ | 0.83 ± 0.15$^f$ | 0.52 ± 0.04$^{de}$ | 1.35 ± 0.19$^f$ | 0.58 ± 0.02$^f$ |
| – | 180 | 0.29 ± 0.13$^c$ | 0.36 ± 0.06$^c$ | 0.52 ± 0.04$^g$ | 0.34 ± 0.04$^e$ | 0.86 ± 0.08$^d$ | 0.42 ± 0.07$^g$ |
| F-value | | 1.98 | 3.67 | 80.40 | 60.97 | 188.47 | 112.16 |
| CV-Value | | 17.87 | 12.38 | 12.22 | 12.89 | 7.69 | 10.51 |

Notes.
Chl a, chlorophyll a; Chl b, chlorophyll b.

**Table 5  Stress tolerance indices calculation based on dry biomass for *P. roxburghii* seedlings.**

| Experiments | Pb levels | | | | | | Cd levels | | | | | |
|---|---|---|---|---|---|---|---|---|---|---|---|---|
| | 30 | 60 | 90 | 120 | 150 | 180 | 30 | 60 | 90 | 120 | 150 | 180 |
| STI% of HE | 109.61 | 91.98 | 77.56 | 62.98 | 54.92 | 44.23 | 112.5 | 89.42 | 81.41 | 67.32 | 57.05 | 46.79 |
| STI% of HtE | 83.77 | 74.03 | 63.69 | 53.34 | 43.20 | 36.30 | 86.31 | 78.09 | 71.39 | 58.21 | 45.43 | 29.81 |
| % change | +25.84 | +17.95 | +13.87 | +9.64 | +11.73 | +7.93 | +26.19 | +11.33 | +10.02 | +9.11 | +11.62 | +16.98 |

Notes.
Key: STI, stress tolerance index; HE, homogeneous environment; HtE, heterogeneous environment.

their subsequent growth. We found that Pb and Cd did not affect the final germination percentage, as all the seeds germinated regardless of metal concentrations. This resilience may be attributed to the hard seed coat of *P. roxburghii*, which protects the embryo and allows selective metal diffusion. The seed coat likely mitigates the toxic effects of metal stress (*Adrees et al., 2015*). Although metal toxicity has been extensively studied in terms of its impact on seed germination (*Datta et al., 2011*), with findings suggesting that heavy metals reduce water absorption or damage the embryo (*El-Rasafi et al., 2016*), our study observed that Pb and Cd concentrations affected the growth of *P. roxburghii* seedlings. *Arduini, Godbold & Onnis (1996)* have reported that the effects of heavy metal toxicity in plants are stunted growth, leaf chlorosis, and modifications in the function of several enzymes in different metabolic pathways. In our research, Pb and Cd impacted *P. roxburghii* seedlings' morphological traits, possibly due to heavy metal toxicity by Pb on root/shoot length and plant growth (*Rahman et al., 2024*). In the existing studies, the root was more sensitive

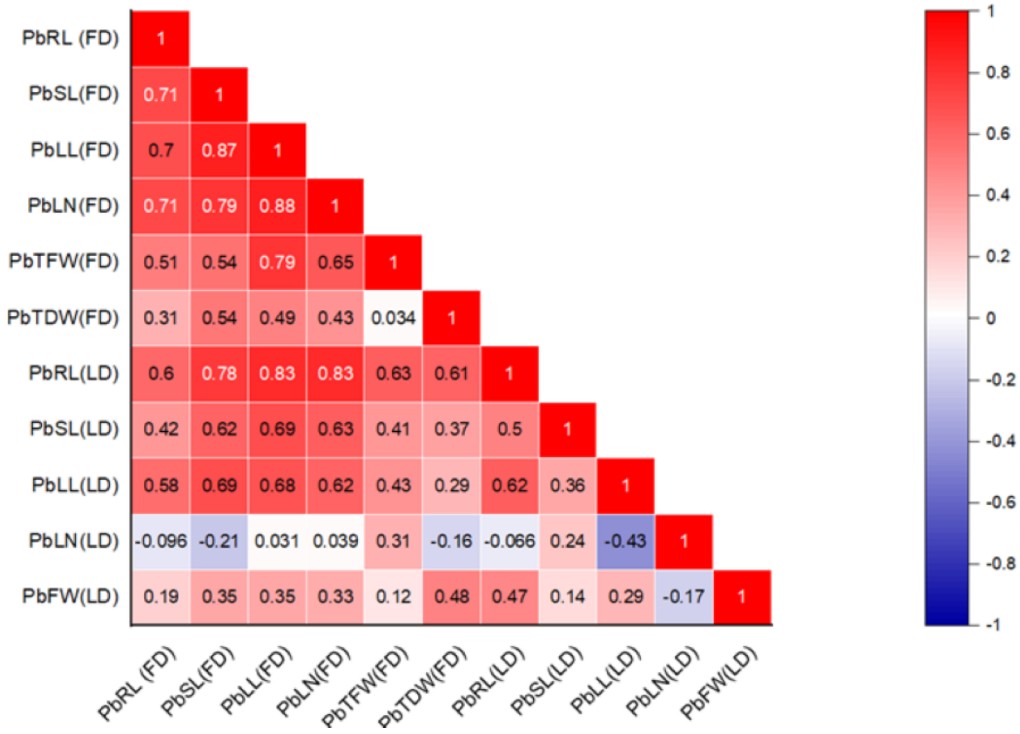

**Figure 1** Correlation heatmap illustrating the relationships among various morphological traits in *P. roxburghii* seedlings exposed to lead (Pb) in both heterogeneous and homogeneous settings (for abbreviations see captions of Tables 1 and 3).

to Cd and Pb than the shoot. With increasing Cd and Pb concentrations, a significant reduction in the root length of *P. roxburghii* was observed. The root is an important plant organ that absorbs nutrients. Under stress conditions, the plant root is the first organ that faces the stress. Plant roots have certain developmental adaptability (*Mohd-Radzman & Drapek, 2023*) and plants can alter their morphology to adapt to environmental stress (*Ru et al., 2022*). The Cd toxicity symptoms are first expressed in plant roots (*Li et al., 2023*). The metal absorption ability of plant roots directly affects their metal enrichment ability. However, leaf length and numbers were less affected by Cd and Pb treatments. *P. roxburghii* leaves are needle-like with a thick cuticle and a waxy coating, which may limit heavy metals uptake and reduce translocation to leaves.

Our study demonstrated that Pb and Cd had a damaging effect on photosynthetic pigments. The decreased chlorophyll content is associated with heavy metal toxicity. Heavy metal toxicity obstructs photosynthetic activity in plants causing reductions in plant height and biomass (*Anwaar et al., 2014*). This decrease in chlorophyll contents is probably due to the inhibition of the reductive steps in the biosynthetic pathways of photosynthetic pigments due to the high redox potential of many metals. Furthermore, the decline in chlorophyll content in plants exposed to heavy metals is believed to be due to inhibiting enzymes such as protochlorophyllide reductase (*Van Assche & Clijsters, 1990*). Cd stress also inhibited plant photosynthesis, thereby deteriorating their photosynthetic production capacity

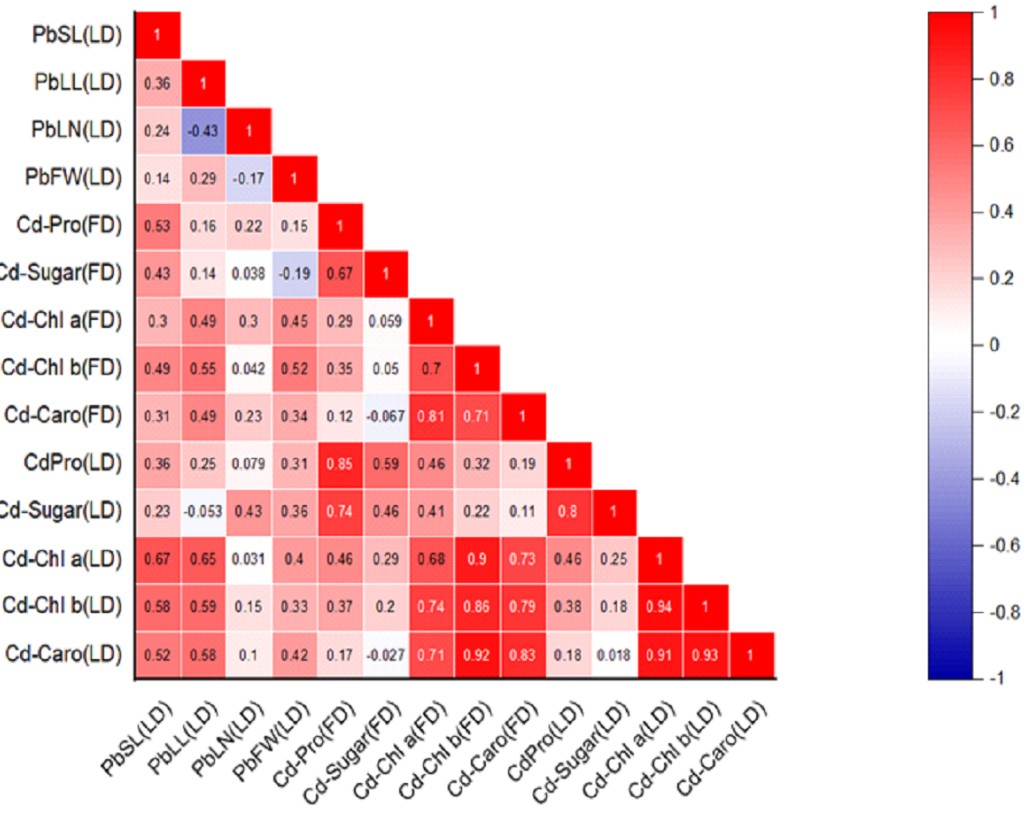

**Figure 2** Relationships among various morphological traits and organic solutes in *P. roxburghii* seedlings exposed to Cd in both heterogeneous and homogeneous settings (for abbreviations see captions of Tables 1–4).

and eventually decreasing their biomass (*Wang et al., 2008*). The morpho-physiological appearance of leaves is highly sensitive to stress conditions, especially to heavy metal stress. It is known that heavy metal toxicity often leads to reduced leaf chlorophyll content, leaf length, greenness, and various other leaf metabolic processes (*Ewais, 1997*; *Seregin & Ivanov, 2001*). Studies on the effects of Pb and Cd on *Brassica juncea* showed a decrease in growth, chlorophyll, and carotenoids (*John et al., 2009*).

The accumulation of osmoprotectants is a physiological and molecular mechanism that supports plants' survival under stress, including heavy metals, by maintaining cellular osmotic balance. Soluble sugars and free proline are low molecular weight osmoprotectants that regulate osmotic potential within plant tissues upon exposure to heavy metal stress. When Pb and Cd stress levels were low or moderate, there was a high accumulation of the osmolytes in the seedlings in both heterogeneous and homogeneous conditioning. Literature studies have revealed that free proline accumulation under stress indicates plant resistance (*Ghosh et al., 2022*). Stress conditions also cause an increase in sugar content due to the breakdown of starch and other sugars (*Dong & Beckles, 2019*). Plants maintain their cell turgor and osmoregulatory mechanisms by accumulating several osmolytes in response to biotic and abiotic stresses (*Anjum et al., 2011*). Several abiotic stresses have

been documented to enhance the production of free proline, soluble sugars, and phenolic contents (*Tan et al., 2006*). Moreover, higher osmolyte accumulation at low and moderate doses suggested the threshold level of heavy metal tolerance in *P. roxburghii*. Proline and sugar are the appropriate solutes for osmotic adaptation, and they accumulate in maximum amounts in response to abiotic and biotic stresses (*Kavi Kishor & Sreenivasulu, 2014*).

The STI is a metric for evaluating a plant species' ability to resist the adverse effects of heavy metal toxicity. Observations from the current study indicate that the growth and biomass of *P. roxburghii* decrease with increasing concentrations of Pb and Cd. This decrease is less obvious in homogeneous environments than in heterogeneous ones. *Tsakou, Roulia & Christodoulakis (2003)* noted that the growth and biomass of *Populus euramericana* were similarly affected by metal treatment in field conditions. *Ansari et al. (2013)* and *Gomes et al. (2013)* also described a similar reduction in plant growth and biomass with increasing As concentrations in the growth medium. The analysis of our results shows that the threshold levels for Cd and Pb tolerance in *P. roxburghii* fall within the range of 90 to 120 mg/L for each metal. This suggests that *P. roxburghii* can withstand Cd and Pb concentrations up to these levels without any notable impact on its morphological traits.

## CONCLUSIONS

In this study, we examined how *P. roxburghii* seedlings respond to heavy metal stress in both nursery and laboratory settings. During their developmental phases, *P. roxburghii* seedlings responded differently to varying levels of Pb and Cd, adjusting their photosynthetic, physiochemical, and growth strategies to adapt to the contaminated environment. By integrating morphological indices with physiochemical analytical data, our findings suggest that these morphological and physiochemical indices can serve as sensitive markers of heavy metal stress in *P. roxburghii* seedlings. Our results conclude that *P. roxburghii* can withstand Cd and Pb concentrations up to 90 to 120 mg/L without any notable impact on its morphological traits. This species can also be utilized to develop green spaces in high-traffic areas of northern Pakistan.

## ACKNOWLEDGEMENTS

The authors extend their gratitude to the Department of Botany, University of Malakand Chakdara Dir (L) Khyber Pakhtunkhwa, Pakistan, the local Bandagai nursery for their invaluable assistance, all of which have played pivotal roles in the successful completion of this research.

### Funding

This work was supported by the Researchers Supporting Project (no. RSP2025R374) King Saud University, Riyadh, Saudi Arabia. The funders had no role in study design, data collection and analysis, decision to publish, or preparation of the manuscript.

## Grant Disclosures

The following grant information was disclosed by the authors:
King Saud University, Riyadh, Saudi Arabia: RSP2025R374.

## Competing Interests

The authors declare there are no competing interests.

## Author Contributions

- Shabana Bibi conceived and designed the experiments, performed the experiments, analyzed the data, prepared figures and/or tables, authored or reviewed drafts of the article, and approved the final draft.
- Tour Jan conceived and designed the experiments, performed the experiments, analyzed the data, prepared figures and/or tables, authored or reviewed drafts of the article, and approved the final draft.
- Nasrullah Khan conceived and designed the experiments, analyzed the data, prepared figures and/or tables, authored or reviewed drafts of the article, and approved the final draft.
- Muhammad Wahab conceived and designed the experiments, analyzed the data, prepared figures and/or tables, authored or reviewed drafts of the article, and approved the final draft.
- Mohammad K. Okla conceived and designed the experiments, analyzed the data, prepared figures and/or tables, authored or reviewed drafts of the article, and approved the final draft.
- Bandar M. Almunqedhi conceived and designed the experiments, analyzed the data, prepared figures and/or tables, authored or reviewed drafts of the article, and approved the final draft.
- Ibrahim A. Saleh conceived and designed the experiments, analyzed the data, prepared figures and/or tables, authored or reviewed drafts of the article, and approved the final draft.
- Yasmeen A. Alwasel conceived and designed the experiments, analyzed the data, prepared figures and/or tables, authored or reviewed drafts of the article, and approved the final draft.
- Saud Alamri conceived and designed the experiments, analyzed the data, prepared figures and/or tables, authored or reviewed drafts of the article, and approved the final draft.
- Hayat Ullah conceived and designed the experiments, analyzed the data, prepared figures and/or tables, authored or reviewed drafts of the article, and approved the final draft.
- Mostafa A. Abdel-Maksoud conceived and designed the experiments, analyzed the data, prepared figures and/or tables, authored or reviewed drafts of the article, and approved the final draft.

## Data Availability

Raw data are provided as a Supplemental File.

## Supplemental Information

Supplemental information for this article can be found online at http://dx.doi.org/10.7717/peerj.19593#supplemental-information.

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
