# Peer review of "Lead and cadmium toxicity effects on the Pinus roxburghii seed germination and early seedling growth in different environments"

_PeerJ, doi:10.7717/peerj.19593_

## Round 0.1 · original submission · Major Revisions

Dear Authors

The manuscript cannot be accepted for publication in its current form. It needs a major revision before publication. The authors are invited to revise the paper considering all the suggestions made by the reviewers. Please note that the requested changes are required for publication.

With Thanks

Reviewer 1 ·

Basic reporting

Thank you for considering me to review the manuscript titled “Lead and cadmium toxicity effect on the Pinus roxburghii seed germination and early seedling growth in different environments". The manuscript addresses a critical and timely topic of heavy metal toxicity. The study provides valuable insights into the physiological and biochemical responses of Pinus roxburghii to lead and cadmium under heterogeneous (nursery) and homogeneous (laboratory) conditions. The dual-environment approach is a notable strength, allowing for a comprehensive understanding of environmental influences on heavy metal stress tolerance. The findings contribute to a broader understanding of forest species responses to environmental stressors and their potential applications in polluted ecosystems. The manuscript is well-structured and supported by robust data, but certain areas require refinement.

Suggestions:
The manuscript is generally well-written and uses clear language. However, minor grammatical errors and phrasing could be refined to ensure it meets high scientific communication standards.

The introduction provides a strong context for the research. Key references are cited, but the authors could expand on recent studies exploring heavy metal tolerance in forest tree species. Moreover, clarifying the rationale for selecting different concentrations of Pb and Cd would be helpful.

The experimental design is appropriate, with clear methods for germination, biochemical analysis, and stress tolerance index calculations. However, clarifying how the environmental differences (heterogeneous vs. homogeneous) impact the outcomes would be helpful.

The results are comprehensive and well-supported by statistical analyses. The use of ANOVA and correlation coefficients is appropriate for interpreting the data. The authors could include more visualizations, such as bar graphs or scatter plots, to summarize trends across treatments.
The figures and tables are clear and relevant, but the legends of Figures 1 and 2 need more detail to stand alone without referring to the text, the abbreviations of studied traits should be explained in the legends.

The discussion addresses the results well but lacks depth in contextualizing findings within existing studies. For example, compare the observed STI trends with other tree species tested under similar conditions. Provide a detailed explanation of the observed differences between heterogeneous and homogeneous environments in terms of seedling physiology. Also, discuss the implications of the observed tolerance thresholds for practical phytoremediation strategies. Moreover, explore the practical implications of the study findings.

The conclusions align well with the data presented. However, some claims, such as the implications for roadside plantations, could be better supported by comparative studies or field validation.

The reference style should be consistent and follow the PeerJ format. For example, journal names are inconsistently abbreviated, such as Environ. Sci. Pollut. Res. (line 339) is abbreviated, while Journal of Agricultural Technology (line 369) is not. Additionally, scientific names should be italicized throughout the manuscript, such as those in lines 349 and 355.

Experimental design

The experimental design is appropriate

Validity of the findings

The findings are supported by robust data.

Reviewer 2 ·

Basic reporting

The article titled "Lead and Cadmium Toxicity Effect on the Pinus roxburghii Seed Germination and Early Seedling Growth in Different Environments" presents valuable insights into the impact of heavy metal toxicity on the germination and early growth of Pinus roxburghii. While the study addresses a significant environmental concern, the manuscript requires major revisions to enhance its clarity, scientific rigor, and overall quality.
While the harmful effects of Pb and Cd on seed germination and seedling growth are highlighted, the physiological mechanisms underpinning these effects (e.g., ion toxicity, oxidative stress pathways) need more depth and clarity.
Several references cited in introduction section are outdated (e.g., Ewais 1997, Henry 2000, Singh et al. 2007). Provide updated references on heavy metal stress in plants to strengthen the scientific basis.
While the introduction touches on various effects of heavy metals, it does not clearly explain why P. roxburghii was chosen as a model species. More emphasis should be placed on the ecological importance, physiological traits, or unique characteristics of P. roxburghii that make it suitable for studying Pb and Cd toxicity.
Include a more detail on how Pb and Cd disrupt plant metabolic and physiological processes (e.g., nutrient uptake, photosynthesis, enzyme activity).
Provide a stronger rationale for studying P. roxburghii, highlighting its ecological or physiological importance.
Remove less relevant information (e.g., human health effects) and focus on plant-specific physiological impacts and the study’s objectives.
Incorporate recent studies to provide a more current understanding of heavy metal stress and phytoremediation strategies.
Materials and Methods: The heavy metal concentrations (0–180 mg/L) are stated but lack justification based on environmental relevance or prior studies. Include references or a rationale for these specific levels.
While some replicates are mentioned (e.g., 3 duplicates of 9 seeds for germination), others (e.g., biochemical indices) are less clear. Standardize and explicitly state replication for all measurements.
The environmental conditions for the nursery (e.g., temperature, humidity, soil properties) and laboratory (e.g., light intensity, temperature) are not described. These are critical for interpreting the results and comparing heterogeneous vs. homogeneous environments.
The use of plastic bags filled with natural soil is unconventional and may introduce variability due to differences in soil composition. Clarify how soil uniformity was ensured and provide details about soil properties (e.g., pH, texture, organic matter).
Add details about the calibration curve and sugar standards used in phenol-sulfuric acid method (Dubois et al., 1956). Add details in the supplementary materials.
Mention the software or version used for statistical analysis. Also mention their settings, and post hoc tests (if any).
In results section, improve grammatical precision and eliminate redundancy. Use standardize terminology for describing trends (e.g., use "increase" and "decrease" instead of ambiguous terms like "trend"). Include brief interpretations of statistical or biological relevance for reported values to clarify their importance in the context of Pb and Cd stress.

In Discussion section many points (e.g., decreased photosynthetic pigments, proline and sugar accumulation, and thresholds of heavy metal tolerance) are repeated, making the discussion redundant in parts. Consolidate overlapping statements and avoid reiterating findings unnecessarily.
While osmoprotectant accumulation is mentioned, the underlying physiological and molecular mechanisms are not discussed. Include potential pathways or mechanisms to provide a more comprehensive explanation of the observed responses.
Highlight species-specific traits that contribute to its resilience or sensitivity to Pb and Cd.
Although the reduction in agronomic traits is noted, the section lacks detailed analysis or hypotheses regarding why certain traits (e.g., number of leaves) were less affected than others. Provide more nuanced insights into differential trait sensitivity.
Some cited references (e.g., Aidid and Okamoto, 1993) are outdated, incorporate findings from recent research to enhance relevance and credibility.
The phrase “threshold level of heavy metals tolerance” is used without quantification or deeper exploration. Clearly define the threshold levels observed in the study and discuss how they compare with those reported for other species or conditions.
The discussion mentions that Cd inhibits photosynthesis but does not detail whether this is due to reduced enzyme activity, stomatal closure, or other specific factors. Expand on the mechanisms by which heavy metals impair photosynthesis, supported by relevant literature.
The conclusion refers to "remarkable development characteristics" and "enhanced metal resistance" without quantifying these traits or comparing them with other plant species. This makes the claims appear overly broad. Specify the extent of resistance observed and compare it to existing data from other species to contextualize the findings.
Some elements of the conclusion (e.g., proline and sugar accumulation, reduced photosynthetic damage) are overly detailed and repeated from the discussion. Focus the conclusion on overarching insights rather than reiterating specific findings.
Avoid repeating detailed observations from the results or discussion. Instead, synthesize the core message of the study.
Highlight how these findings can be applied in real-world scenarios or in advancing scientific understanding of heavy metal stress in plants.
Introduce potential areas for future research, such as exploring molecular mechanisms or testing responses to other heavy metals or environmental conditions.
Specific comments:
Several phrases are vague or need further explanation. For example: "Pb and Cd can be assiduously absorbed by plant roots" (Line 45): The term "assiduously" is uncommon in scientific writing and could be replaced with "readily" or "actively."
Some statements, like the impact of Pb and Cd on germination and growth (Lines 44–49), are supported by multiple references, but their relevance to P. roxburghii specifically is not clear.
The introduction section covers the general toxicity of Pb and Cd on plants and humans but could better focus on the physiological and biochemical aspects specific to plants. For example, the discussion on health issues in humans (Lines 46-47) could be abbreviated or omitted to maintain a plant-specific focus.
"Oxidative stress through free radicals' production" (Line 55): The mechanism of free radical generation could be described in more detail, such as the role of reactive oxygen species (ROS) and their impact on cellular structures.
Redundant information exists, such as repeated mentions of oxidative stress and seed germination (Lines 52-57 and 66-69).
The introduction lacks a clear progression from general heavy metal stress to the specific study objective. For example, the transition to P. roxburghii as the focus species (Lines 77–86) feels abrupt. It would be more effective to gradually introduce the relevance of P. roxburghii in the context of metal stress and phytoremediation.
The inclusion of references to phytoremediation (Lines 60–72) is useful but somewhat disjointed from the rest of the introduction. It should be more tightly integrated with the discussion on the ecological and physiological aspects of heavy metal stress.
The last two sentences (Lines 85–89) succinctly state the study objective but lack a clear hypothesis or research question, which would help frame the study more effectively.
The metals were applied “four times at 12-day intervals” (Line 98), but the rationale behind this frequency is missing. Was it based on metal uptake kinetics or plant tolerance levels?
The significance level of P < 0.05 is inaccurately described (“probability of occurrence of less than 50%”). It should be corrected to state that results are considered significant if the likelihood of the null hypothesis being true is less than 5%.
Ensure that the table captions are revised to provide a precise and comprehensive description of the data, variables, and parameters presented, ensuring alignment with the content detailed in each table.

Experimental design

Experimental design requires further clarification and a more rigorous description to address the gaps highlighted in the basic reporting section.

Validity of the findings

The results presented in Table 2 and table 4 appear to lack scientific accuracy. Specifically, the reported chlorophyll a:b ratio contradicts established physiological principles. Similarly, the total chlorophyll content values are inconsistent with the reported chlorophyll a and b values. Additionally, proline, a well-documented stress indicator, is reported at unexpectedly low levels under maximum lead stress in Table 2, recheck it. These discrepancies warrant a thorough re-evaluation and re-calculation of the data to ensure the scientific rigor and credibility of the study.

Reviewer 3 ·

Basic reporting

The manuscript entitled "Lead and cadmium toxicity effect on the Pinus roxburghii seed germination and early seedling growth in different environments" is a significant study providing the insights into the adaptive responses of Pinus roxburghii seedlings to lead (Pb) and cadmium (Cd) stress, highlighting its potential use in reforestation and phytoremediation of heavy metal-contaminated environments. The manuscript is well-structured with clear sections outlining the introduction, methods, results and discussion. However certain sections have queries that need to be addressed properly, for the improvement of the manuscript.

The introduction extensively reviews the toxic effects of lead and cadmium in general, however the information about specific relevance of this species towards this study is inadequate. Make a connection between significance of this species and its susceptibility to heavy metals.
Although the objectives of study are clear, research gap and novelty is not clearly described. Authors could emphasize gaps in knowledge about the specific impacts of these metals in heterogeneous vs homogeneous environments, as it seems to be a novel aspect of the study.

Experimental design

The experimental setup lacks the information about environmental parameters of both environments where the experiments were conducted. The variables like temperature, humidity, soil composition etc can significantly influence plant responses to heavy metal stress and should be monitored and reported.

For data analysis, using one way ANOVA may not sufficiently represent the complexity of the data.
Given the two distinct environments and the potential for interactions between treatment and environmental conditions, a two way ANOVA might have provided deeper insights into the effects of Pb and Cd under different conditions.

The description of the significance level as "a probability of occurrence of less than 50%" is misleading. It needs clarification.

Also add the details about software used for statistical analyses.

Validity of the findings

In results section, significance is not represented in the form of p values.

Make consistency in using terms of homogenous vs laboratory, heterogeneous vs field or nursery conditions, in sub headings especially. Similarly the terms; morphology, morphometry and growth.

Results for leaf number variation is mentioned in the section. However, this trait is not mentioned in methodology section.

The discussion section addresses the findings well. However it needs a connection between findings and study's objectives, supported by recent and relevant literature.
The study investigated heterogeneous and homogeneous environments, yet the discussion does not differentiate between the impacts observed in these environments.

In conclusion section, adding ecological or practical implications of the findings like the potential use of P. roxburghii in phytoremediation or ecosystem restoration in polluted areas would enhance the applied significance of the study.

Additional comments

Check the format of references in the list and in-text also. Make it consistent.
Replace very old references of discussion with recent one.
Italicize the scientific names of plants.

Annotated reviews are not available for download in order to protect the identity of reviewers who chose to remain anonymous.

---

## Round 0.2 · Major Revisions

Dear Authors
The manuscript cannot be accepted for publication in its current form. It needs a major revision before publication. The authors are invited to revise the paper, considering all the suggestions made by the reviewers. Please note that the requested changes are required for publication.
With Thanks

Reviewer 1 ·

Basic reporting

The author has addressed all the previously mentioned comments

Experimental design

The experimental design is appropriate

Validity of the findings

The findings are supported by robust data.

Reviewer 2 ·

Basic reporting

The authors have made efforts to incorporate some of the suggested improvements from the previous revision. However, several critical issues remain unresolved and must be addressedok.
Please ensure that the statistical notation is correctly formatted throughout the manuscript. Specifically, the symbol p indicating the probability value should be italicized consistently, following standard scientific writing conventions.
Table 2 and Table 4 continue to present ambiguous and potentially inaccurate data regarding chlorophyll content. It is scientifically implausible for the total chlorophyll content to be lower than chlorophyll a, as total chlorophyll is the sum of chlorophyll a and b. The authors are requested to carefully recheck the calculations and provide clarification or correction where necessary.
A major concern remains regarding data transparency. The raw data for the physiological and biochemical parameters measured in the study have not been provided. This omission significantly limits the ability to verify and interpret the results. The authors are strongly advised to include the raw data in supplementary material.

Experimental design

Ok

Validity of the findings

Validity of the study is compromised due to vague data of chlorophyll content and unavailability of raw data regarding physiological and biochemical tests.

Reviewer 3 ·

Basic reporting

Authors have addressed well all the queries raised by the reviewers. The revised version of manuscript now well describes the objectives and clarifies the research gap of the study. This revised version of manuscript can now be considered further.
However, authors need to recheck the grammar throughout the manuscript especially within the edited portion. At some places, change to past tense is required.
Replace the word "morphologic" with "morphological".
Recheck the in-text citation format and botanical nomenclature format.

Experimental design

Experimental design and statistical analyses have been described well with sufficient detail.

Validity of the findings

All the mentioned shortcomings have been fulfilled in results and discussion sections.

---

## Round 0.3 · accepted · Accept

Dear Authors,

I am pleased to inform you that the manuscript has improved after the last revision and can be accepted for publication.

Congratulations on accepting your manuscript, and thank you for your interest in submitting your work to PeerJ.

With Thanks

Reviewer 2 ·

Basic reporting

I am pleased to see authors have addressed most of the comments raised during previous round of revision. Now the article can be accepted for publication in your esteemed journal.

Experimental design

The manuscript is solid in scientific content and experimental design.

Validity of the findings

Findings seems valid.

Reviewer 3 ·

Basic reporting

The revised version of manuscript has been improved by following the reviewers' comments. Authors have tried well to address all the queries raised by the reviewers. The manuscript can now be processed further.

Experimental design

Experimental design is appropriate and methods have been well described.

Validity of the findings

The findings of research have been well presented.